# rePIRL: Learn PRM with Inverse RL for LLM Reasoning

## Abstract

Providing process rewards has been widely used in deep reinforcement learning to improve training efficiency, reduce variance, and prevent reward hacking. In LLM reasoning, existing works also explore various solutions for learning effective process reward models (PRM) with or without the help of an expert policy. However, existing methods either rely on strong assumptions about the expert policies (*e.g.,* requiring their reward functions) or suffer intrinsic limitations (*e.g.,* entropy collapse), resulting in weak PRMs or limited generalizability. In this paper, we introduce rePIRL, an inverse RL-inspired framework that learns effective PRMs with minimal assumptions about expert policies. Specifically, we design a dual learning process that updates the policy and the PRM interchangeably. Our learning algorithm has customized techniques to address the challenges of scaling traditional inverse RL to LLMs. We theoretically show that our proposed learning framework can unify both online and offline PRM learning methods with additional assumptions, justifying that rePIRL can learn PRMs with minimal assumptions. Empirical evaluations on standardized math and coding reasoning datasets demonstrate the effectiveness of rePIRL over existing PRM learning methods. Our ablation studies further show the effectiveness of our key designs.

## 1 Introduction

Intermediate rewards have demonstrated their effectiveness in training deep reinforcement learning agents for various applications, *e.g.,* simulated games (Ng et al., 1999), Go (Silver et al., 2016), and Poker games (Brown & Sandholm, 2019). For example, in MuJoCo environments, intermediate rewards are introduced to guide the robot to progressively learn to stand, walk, and finish its corresponding tasks (Heess et al., 2017). Intermediate rewards are also helpful in accelerating learning convergences (Wu et al., 2023), reducing variances (Laud, 2004), and preventing reward hacking (Ziegler et al., 2022). Inspired by these successes on DRL agents, recent works also explore training process reward models (PRMs) to provide token-level intermediate reward during LLM reasoning. At a high level, these works can be categorized as off-line methods (Uesato et al., 2022; Lightman et al., 2023; Wu et al., 2023; Xu et al., 2024; Yoon et al., 2024), which learn a PRM based on a given expert policy, or on-line methods (Wang et al., 2023; Zhao et al., 2025b; Cui et al., 2025a), which learn a PRM based on internal signals of the reasoning model or the final outcome reward. However, existing PRM methods either add strong assumptions about the expert policies and reward modeling or suffer intrinsic limitations.

More specifically, within offline methods, supervised PRM methods (Uesato et al., 2022; Lightman et al., 2023) or DPO-type of works (Yuan et al., 2024b; Cui et al., 2025a) (although do not explicitly learn a PRM) require access to token-level annotations or preference labels of expert trajectories. Monte Carlo tree search (MCTS)-based methods require running the expert policy and recording the final outcome reward, which is time-consuming and no longer works if the expert policy is not available. For online methods, PRIME (Cui et al., 2025a) learns a PRM based on the outcome reward of the current policy, which only works under strict assumptions about the reward functions and the policy model under training (see Section 3.3). Intrinsic reward methods (Zhao et al., 2025b) explore using the reasoning model's response entropy or confidence as the process reward, but suffer from the entropy collapse issue, degrading performance in later training stages (Cui et al., 2025b).

Table 1: Comparison of rePIRL and SOTA PRM methods. ✓ means the corresponding method does not suffer this issue, ✗ otherwise. "Supervised" refers to the methods that learn PRMs from supervised labels and "Intrinsic" refers to the methods that learn PRMs from model confidence or entropy. Assumptions refer to the method-specific assumptions discussed in Section 3.3.

|  | Supervised | MCTS | DPO | PRIME | Intrinsic | rePIRL |
|---|---|---|---|---|---|---|
| Token annotation/label | ✗ | ✓ | ✗ | ✓ | ✓ | ✓ |
| Expert reward | ✗ | ✗ | ✗ | ✓ | ✓ | ✓ |
| Expert policy | ✗ | ✗ | ✗ | ✓ | ✓ | ✓ |
| Assumptions | ✓ | ✗ | ✗ | ✗ | ✓ | ✓ |
| Entropy collapse | ✓ | ✓ | ✓ | ✓ | ✗ | ✓ |

In this paper, we propose rePIRL, a novel learning framework that learn effective PRMs for LLM reasoning from only the expert trajectories without access to process reward annotations, preference labels, and the expert policy. rePIRL also do not require the strong assumptions of the online method, PRIME, and do not suffer the entropy collapse issue of the intrinsic reward methods. At a high level, we borrow the idea from inverse reinforcement learning (IRL), which is proposed to learn a reward function from given expert trajectories, using which one can learn a policy as optimal as the expert policy. We start with modeling multi-step LLM reasoning as a token-level MDP and design the learning objective function for our PRM following the classical IRL framework, *i.e.,* defining a hidden variable followed by energy-based modeling. To make sure the objective function is tractable for LLMs, we then apply the gradient trick to get rid of the partition function and apply importance sampling to avoid sampling from the expert policy (which is not available). Following this process, we design our learning algorithm to learn PRM and policy interchangeably, where the policy loss is the maximum entropy RL loss, which can be solved by PPO (Schulman et al., 2017) or its variants (RLOO (Lu et al., 2023), GRPO (Shao et al., 2024)). After deriving our learning algorithm, we further conduct rigorous theoretical analysis to integrate representative online and offline PRM learning methods into our framework: PRIME (Cui et al., 2025a), Math-Shepherd (MCTS) (Wang et al., 2023), DPO (Rafailov et al., 2024), and DQO (Ji et al., 2024). We show that these methods can be integrated into our learning framework with additional assumptions on top of rePIRL, justifying that *rePIRL is the PRM learning method with the minimal required assumptions*.

We apply rePIRL to the Qwen-2.5-3B model and compare it with behavioral cloning, PRIME (Cui et al., 2025a), and MCTS-based PRM training method (Wang et al., 2023) on 7 standard math and coding reasoning benchmarks. The experiment results demonstrate the superiority of our method compared to offline and online PRM baselines. It consistently outperforms these methods, obtaining an average of 33.5% accuracy on mathematical reasoning tasks and 27.7% accuracy on coding tasks. We conduct an ablation study to show that rePIRL works best with RLOO for policy update, compared to PPO and GRPO, as well as the testing phase scaling of our learned policies. Finally, we show that a policy trained solely with our learned PRM without any outcome signal can reach competitive performance on selected benchmarks, further validating the effectiveness of our reward modeling. To the best of our knowledge, rePIRL is *the first work that generalizes IRL for LLM reasoning*. It is also *the first work that unified the existing PRM learning methods into the same learning framework*.

Our key contributions are: 1) We propose rePIRL, a novel PRM learning framework, inspired by classical IRL, together with a scalable learning algorithm for both PRM and policy training. 2) We proposed a unified post-training framework that includes SOTA reasoning methods, especially DPO, DQO, PRIME, and MCTS. For each method, we analyzed its underlying assumptions and connections to our framework. Through this analysis, we establish a unified framework, a better understanding of different reasoning methods, and a conclusion that rePIRL is a generalized PRM learning method that operates under the minimal necessary assumptions. 3) We empirically demonstrate the clear advantage of rePIRL over baseline methods and systematically validate the effectiveness of our key design choices.

## 2 RELATED WORK

There have been a number of works on LLM reasoning that learn process reward models with or without expert trajectories or policies. As shown in Table 1, these methods generally adopt stronger

assumptions than ours, and can be integrated into our unified framework outlined in Section 3. We group the prior research by whether it requires expert policies.

**Methods require expert policies.** Early works in this category train a process reward model from data with annotated token-level or step-level rewards using supervised learning losses (Uesato et al., 2022; Wu et al., 2023; Xu et al., 2024; Yoon et al., 2024; Ma et al., 2023; Lightman et al., 2023). The annotations are given by human experts or other well-trained LLMs. Without assessing the token-level rewards, another line of work proposes to run the expert policy-based Markov chain tree search and generate trajectories with process reward labels (Zhao et al., 2025a; Lee et al., 2024; Zhang et al., 2024b; Wang et al., 2023; Zhang et al., 2024a; Xiong et al., 2025b). Here, the label for each intermediate reasoning step is assigned based on the final outcomes of the Markov samples drawn from this step. These methods require access to the expert policy. More importantly, the MCTS process is time-consuming and computationally expensive, especially when we want to calculate token-level rewards.

Note that directed policy optimization (DPO) and its follow-ups (Rafailov et al., 2023; Richemond et al., 2024; Ethayarajh et al., 2024) directly learn new policies from requiring expert trajectories without explicitly modeling the reward function. DPO has the bandit assumption, *i.e.,* the model cannot consider multi-step reasoning. Recent offline RL works extend DPO to multi-step reasoning by leveraging a trajectory-level Bradley-Terry preference model (Rafailov et al., 2024; Zeng et al., 2024; Meng et al., 2024) or leveraging the relations between the optimal policy model and value functions in maximum entropy RL (Ji et al., 2024; Wang et al., 2024; Zhong et al., 2024). As these works do not explicitly learn a reward function, we do not consider them as baselines in our work.

**Methods do not require external signals.** Yuan et al. (2024a) proposes a specific format for the reward function, which can be used as both process and outcome rewards. Cui et al. (2025a) (PRIME) leverages this reward format and proposes to learn a PRM based on outcome reward and policy interchangeability. As we will show in Section 3, this specific format only holds under certain assumptions (Section 3.3). Other works explore using model internal confidence or entropy as process reward (Zhao et al., 2025b; Li et al., 2025a). While such approaches can be beneficial during the early stages of fine-tuning, they often suffer from entropy collapse, which can degrade performance in later training stages (Cui et al., 2025b).

Note that we do not consider the works that learn reasoning models through supervised imitation learning (distillation) (Singh et al., 2023; Dong et al., 2023; Hao et al., 2024; Muennighoff et al., 2025; Li et al., 2025b; Ye et al., 2025; Xia et al., 2025; Kang et al., 2025; Wang et al., 2025; Hao et al., 2024; Cui et al., 2025c; Ma et al., 2025). Some recent methods improve the online RL algorithms (*e.g.,* PPO) for LLM reasoning (Xiong et al., 2023; 2025a; Arora & Zanette, 2025; Guo et al., 2025; Luong et al., 2024; Yeo et al., 2025; Yu et al., 2025; Yuan et al., 2025; Hu, 2025; Li et al., 2023b; Williams, 1992; Ahmadian et al., 2024; Schulman et al., 2017), which can be used in our methods to update the policy model. Finally, there are some studies of inverse RL before the emergence of LLMs (Ziebart et al., 2008; Finn et al., 2016b). There is also some work about inverse RL for alignment (Sun & van der Schaar, 2024; Zeng et al., 2025; Li et al., 2024b). These works mostly are designed for alignment rather than reasoning. As such, the fundamental model is different. As stated in their papers, alignment mainly optimizes over the entire trajectories, which fundamentally follows a bandit assumption. However, reasoning requires modeling the problem as a real MDP, making the problem much more challenging.

# 3 KEY TECHNIQUE

## 3.1 PROBLEM SETUP

Given a reasoning LLM denoted by $f$, which takes as input a prompt $\mathbf{x}$ and generates the output tokens autoregressively, where each token is denoted as $y_i$. We define the generation process as a token-level Markov decision process (MDP): $\mathcal{M} = \{\mathcal{S}, \mathcal{A}, T, R, \gamma\}$, where state $s_t \in \mathcal{S}$ is the current output token $s_t = [\mathbf{x}, y_0, ..., y_{t-1}]$. $\mathcal{A}$ is the vocabulary size, and $a_t = y_t$ is the LLM's current output token at time/position $t$. As the next state is obtained by appending the newly generated token to the current output, we have a deterministic state transition function (T). $R(s_t, a_t)$ is the reward for each generated token. For simplicity, we omit the discount factor $\gamma$. Given a set of trajectories $\mathcal{D}$ sampled from an expert policy $\pi_{\text{ref}}$, where each trajectory $\tau = \{(s_t, a_t)\}_{t=1:T}$. The expert policy is

trained from a certain reward function $r_{\text{ref}}(s, a)$. As shown in Table 1, we consider the setup with the fewest assumptions, where we do not assume accessing the reference policy $\pi_{\text{ref}}$ and its reward function $r_{\text{ref}}(s, a)$. For the expert trajectories, we do not require the token-level reward or preference labels. Our *goal* is to recover a reward function from the same family as the reference reward function $r_{\text{ref}}(s, a)$ and recover a policy that is as optimal as possible under the recovered reward model.

First, having the fewest assumptions reduces the thresholds of applying our proposed method, making it much more generalizable. For example, obtaining token-level reward labels requires either substantial human effort or intensive computational cost to conduct MCTS. Our method can operate without such additional efforts and other assumptions. Besides, because our method does not assume access to an expert policy, it can learn from trajectories sampled from experts that do not have an analytic policy function (*e.g.,* humans). Although it is possible to directly learn a nearly optimal policy from the expert trajectories without explicitly learn the reward function (Wang et al., 2024), we believe it is still necessary to recover the reward function in many scenarios. This is because the learned reward function can be used together with other reward functions for further training the policy to learn new capabilities and improve its end-to-end performance. For example, if the reference policy often produces long reasoning processes with redundant steps (*e.g.,* DeepSeek-r1), we can add a reward that penalizes the long reasoning chain. By using it together with the recovered reward, we can potentially train policies that can produce concise reasoning chains without harming their final performance. Another use case is when the expert policy and the learned policy cannot handle a specific type of query well (could be unseen questions), we can use the recovered reward and a reward function designed specifically for those queries to train the policy such that it can better handle these queries while preserving its general performance.

### 3.2 rePIRL Learning Framework

**Technical rationale.** We borrow the idea from classical inverse reinforcement learning, as these techniques are designed for learning a reward function from given expert trajectories without the reward annotations. Following this idea, a straightforward solution would be using an inverse RL method to learn a reward function from the expert trajectories and then train a policy based on the learned reward function. However, the traditional inverse RL problem requires estimating a partition function $z$, which involves traversing all the possible states and actions (Ziebart et al., 2008). This is intractable for LLM models, which have a large action and almost infinite state space.

An approximate solution is to train a policy model together with the reward function and approximate the partition function using the trajectories sampled from the trained policy model. This method avoids the intractable operations in the traditional inverse RL models and has been demonstrated to be effective for shallow neural network policies in robotics (Finn et al., 2016b). In this work, we generalize this solution to LLM multi-step reasoning and propose a unified reward and policy learning framework.

**Objective function construction.** We define the reward function as $r_\phi(s_t, a_t)$, parametrized by $\phi$ and the policy network as $\pi_\theta(a_t|s_t)$. We then define a hidden variable $o_t = 0/1$. $o_t = 1$ means the action $a_t$ in an expert trajectory is sampled from the expert policy rather than a non-optimal one, and $o_t = 0$ otherwise. We define $p(o_t|s_t, a_t) \propto \exp(r_\phi(s_t, a_t))$, an energy-based function (Finn et al., 2016a). Here, $p(\tau|o_{1:T})$ is the probability of $\tau$ being sampled from the expert policy.

Reward function. We can learn the reward function from the following MLE loss: $\mathcal{J}(\phi) = \mathbb{E}_{\tau \sim \mathcal{D}}[\log p(\tau|o_{1:T})]$. That is, given a set of trajectories sampled from the expert policy, we want to maximize their likelihood by learning an optimal reward function.

$$\max_\phi \mathcal{J}(\phi) = \max_\phi \mathbb{E}_{\tau \sim \mathcal{D}}[\log p(\tau|o_{1:T})] = \max_\phi \frac{1}{|\mathcal{D}|} \sum_i \sum_t r_\phi(s_{i,t}, a_{i,t}) - \log z(\phi), \quad (1)$$

where $s_{i,t}$ is the $t$-th state in the $i$-th trajectory ($\tau_i$). $z(\phi) = \int p(\tau) \exp(r_\phi(\tau)) d\tau$ is the partition function. The gradient of the loss function w.r.t. the reward function parameter is

$$\nabla_\phi \mathcal{J} = \mathbb{E}_{\tau \sim \mathcal{D}}[\nabla_\phi r_\phi(\tau)] - \mathbb{E}_{\tau \sim p(\tau|o_{1:T})}[\nabla_\phi r_\phi(\tau)], \quad (2)$$

where $p(\tau|o_{1:T}) = \frac{p(\tau) \exp(r_\phi)}{z(\phi)}$, which is the soft optimal policy under the current reward. Estimating $p(\tau|o_{1:T})$ requires computing the forward and backward pass $\mu_t(s_t, a_t) \approx \beta(s_t, a_t)\alpha(s_t)$ for all time

steps (Ziebart et al., 2008). For the token-level MDP, $\beta(s_t) = \mathbb{E}_{a_t \sim p(a_t|s_t)}[\exp(r_\phi(s_t, a_t))\beta(s_{t+1})]$ and $\alpha(s_t) = \sum_{a_{t-1}} \exp(r_\phi(s_{t-1}, a_{t-1}))\alpha(s_{t-1})$. Because the state and action spaces are enormous and the reasoning models can span many time steps, computing $\alpha$ and $\beta$ becomes intractable.

To resolve this challenge, we propose to sample trajectories $\bar{\tau}$ from our defined policy $\pi_\theta$ rather than the $p(\tau|o_{1:T})$. The gradient in Eqn. (2) is then turned into

$$\nabla_\phi \mathcal{J} = \mathbb{E}_{\tau \sim \mathcal{D}}[\nabla_\phi r_\phi(\tau)] - \mathbb{E}_{\bar{\tau} \sim \pi_\theta(a_t|s_t)}[\frac{p(\bar{\tau}|o_{1:T})}{\pi_\theta(\bar{\tau})} \nabla_\phi \, r_\phi(\bar{\tau})]$$
$$= \frac{1}{N} \sum_i \nabla_\phi r_\phi(\tau_i) - \frac{1}{\sum_j w_j} \sum_j w_j \, \nabla_\phi \, r_\phi(\bar{\tau}_j) \,, \tag{3}$$

where $\nabla_\phi r_\phi(\tau_i) = \sum_t \nabla_\phi r_\phi(s_{i,t}, a_{i,t})$.

$$w_j = \frac{p(s_1) \prod_t p(s_{t+1}|s_t, a_t) \exp(r_\phi(s_t, a_t))}{p(s_1) \prod_t p(s_{t+1}|s_t, a_t)\pi_\theta(a_t|s_t)} = \frac{\exp(\sum_t r_\phi(s_t, a_t))}{\prod_t \pi_\theta(a_t|s_t)} = \frac{\exp(r_\phi(\bar{\tau}_j))}{\pi_\theta(\bar{\tau}_j)} \tag{4}$$

is the importance weight. Instead of solving the optimal policy under the current reward function, we sample from a sub-optimal policy $\pi_\theta$ and mitigate the sample bias with importance sampling. The more we optimize the policy $\pi_\theta$, the closer the importance weight will go to 1. The final objective function for $r_\phi$ can be written as

$$\max E_{\tau \sim \mathcal{D}}[r_\phi(\tau))] - \log(E_{\bar{\tau} \sim \pi_\theta}[\frac{\exp(r_\phi(\bar{\tau}))}{\pi_\theta(\bar{\tau})}]) \,. \tag{5}$$

**Policy model.** At any time during the training, we can update the policy network by the common online RL objective, *i.e.,* maximize the expected total reward $\mathbb{E}_{\bar{\tau} \sim \pi_\theta}[r_\phi(\bar{\tau})]$. However, the optimal policy under this objective function will be a deterministic policy that only takes the action maximizing the value function at each state $a_t = \arg\max Q(s_t, a_t)$. For LLMs, this may cause the degeneration or model collapse issue. To resolve this issue, we propose to learn the policy model via the maximum entropy RL objective function (Haarnoja et al., 2018).

$$\max E_{\bar{\tau} \sim \pi_\theta}[r_\phi(\bar{\tau})] + \beta \mathcal{H}(\pi_\theta(\tau)) = E_{\bar{\tau} \sim \pi_\theta}[r_\phi(\bar{\tau})] + \beta E_{\bar{\tau} \sim \pi_\theta}[\log \pi_\theta(\bar{\tau})] \,. \tag{6}$$

**Learning algorithm.** Algorithm 1 shows our proposed learning algorithm for the reward function and the policy model.

## 3.3 INTEGRATE SOTA METHODS INTO OUR FRAMEWORK

In this section, we discuss how existing offline RL and PRM learning methods can be intergated into our proposed framework by adding assumptions with different strengths. We mainly discuss two representative methods listed in Table 1: Math-Shepherd (Wang et al., 2023), which learns a reward function sololy, and PRIME (Cui et al., 2025a), which learns a reward function together with the policy model, as well as two methods that learn a policy from the expert trajectories without explicitly learning a reward function: DPO (Rafailov et al., 2024) and DQO (Ji et al., 2024).

First, we present the following theorem.

**Theorem 1** *The optimial solution for the maximize entropy RL objective in Eqn. (6) is $\pi_*(a|s) = exp(\frac{1}{\beta}[Q^*(s,a) - V^*(s)])$, where $V^\pi(s_t) = \mathbb{E}_{a_t \sim \pi(\cdot|s_t)}[r(s_t, a_t) + \gamma V^\pi(s_{t+1})] + \beta \mathcal{H}(\pi(\cdot|s_t))$ is the soft state-value function and $Q^\pi(s_t, a_t) = r(s_t, a_t) + \gamma V^\pi(s_{t+1})$ is the soft action-value function.*

Theorem 1 serves as the foundation for our theoretical analysis. We note that this theorem is an established result and not a novel contribution of this work. Our novelty is the analysis of different methods' connection to our framework and their assumptions, as follows.

**Connection to DPO.** As shown in Table 2, when being used to multi-step LLM reasoning, DPO assumes the availability of trajectory preference labels and a reference policy where the offline trajectories are sampled from. Given that DPO does not explicitly learn the reward function, it only has one learning objective function for the policy model. The DPO objective is constructed based on the following relations $\pi_*(a|s) = \frac{1}{Z(s)}\pi_{\text{ref}}(a|s)\exp(\frac{1}{\beta}r(a, s))$ The following proposition states this key relations and our policy model objective in Eqn. 6.

---

**Algorithm 1** rePIRL Algorithm for one iteration

---

**Require:** Policy model $\pi_\theta$, token-level process reward model $\pi_\phi$, outcome reward model $r_o$, training set $\mathcal{D}$, rollout number $n$, expert rollouts $\tau_\mathcal{D}$ for training set $\mathcal{D}$.

 1: **for** Batch $\mathcal{B}$ in $\mathcal{D}$ **do**
 2:     $\hat{\tau}_\mathcal{B} = \pi_\theta(\mathcal{B}, n)$ {Generating $n$ policy rollouts for each question in the batch: $\hat{\tau}_\mathcal{B}$}
 3:     Fetch expert demonstration $\tau_\mathcal{B}$ w.r.t batch $\mathcal{B}$
 4:     Combine policy rollouts and expert rollout
 5:     Calculating the outcome reward $r_o(\tau)$ for each rollout $\tau$
 6:     Calculating the average process reward for each rollout: $\bar{r}_\phi(\tau) = \frac{1}{|\tau|} \sum r_\phi(\tau)$
 7:     For each rollout, if the outcome reward $r_o(\tau)$ is 0, calculate the rollout importance score $w$:

$$w = \frac{\exp(r_\phi(\tau))}{\pi_\theta(\tau)}$$

 8:     Calculate the loss for the process reward model:

$$\mathcal{L} = \frac{1}{\sum w_i} \sum_{\hat{\tau} \in \hat{\tau}_\mathcal{B}} w_i \bar{r}_\phi(\hat{\tau}) - \frac{1}{|\mathcal{B}| * n} \sum_{\tau \in \tau_\mathcal{B}} \bar{r}_\phi(\tau)$$

 9:     Update the reward model according to $\mathcal{L}$
10:     Calculate the process reward score for every policy rollout $\hat{\tau}$ in $\hat{\tau}_\mathcal{B}$
11:     Update the policy model $\pi_\theta$ using RLOO algorithm
12: **end for**

---

**Proposition 1** *When define the reward function as $r'(a, s) = r(a, s) + \beta \log \pi_{ref}$ and the bandit assumption. The optimal solution of the maximum entropy RL is $\pi_*(a|s) = \frac{1}{Z(s)} \pi_{ref}(a|s) exp(\frac{1}{\beta} r(a, s))$.*

This proposition states that the policy learning objective of DPO is a special case of our policy learning objective. The availability of preference label enables DPO to directly learn the policy model using an MLE loss based on the Bradley-Terry model (Bradley & Terry, 1952), eliminating the need to estimate the partition function. However, due to the lack of the partition function, we cannot direct obtain the reward function from a learned policy as $r(s, a) = \beta \log \frac{\pi}{\pi_{ref}} + \beta log Z$. Note that $\log \frac{\pi}{\pi_{ref}}$ is consistent with $r(s, a)$ only under the Bradley-Terry model, however, when being used for policy training, it is biased without the partition function. In summary, *under a specific reward function, DPO optimizes the same policy objective as us. It does not need to estimate the reward function and the partition function thanks to the preference label.*

**Connection to DQO.** DQO assumes the access to token-level reward annotation and design their policy objective functions accordingly.

**Proposition 2** *The objective function of DQO is derived from our policy model objective based on the relation in Theroem 1.*

With proposition 2, we can state that DQO *also optimizes the same maximum entropy RL for the policy model*. With the token-level reward annotations, DQO can approximate the valuate function via TD learning without learning the reward function.

**Connection to Math-shepherd.** Math-shepherd and follow-up works apply MCTS to obtain the process reward annotation and then leverage an MLE (cross-entropy) loss to learn a reward function. The MCTS process is computationally cost and the resulted process reward is a distribution of the final reward and cannot capture the intermediate rewards. In LLM reasoning, the intermediate reward could be the quality and conciseness of intermediate reasoning steps. Besides, it requires accessing and executing the expert policy. Here, we show that our reward objective in Eqn. (5) can also be modeled as a cross-entropy loss.

**Proposition 3** *The objective function in Eqn. (5) is equivalent to the following cross-entropy loss.* $\mathbb{E}_{\tau \sim \mathcal{D}}[\log D(\tau)] + \mathbb{E}_{\tau \sim \pi_\theta}[\log(1 - D(\tau))]$ *where $D = \frac{\tilde{p}}{\tilde{p} + \pi_\theta}$ and $\tilde{p} = \frac{1}{z} \exp(r_\phi(\cdot))$.*

**Connection to PRIME.** Although PRIME is an online method (*i.e.,* does not require expert trajectories), it also learns a reward function together with the policy network. Here we show that PRIME can also be integrated into our framework under certain assumptions. First, PRIME leverages the implicit reward, *i.e.,* model the reward as $r = \log \frac{\pi_\theta}{\pi_{\text{ref}}}$ and use it as the PRM and ORM at the same time. The following proposition states the underline assumptions of having this relationship.

**Proposition 4** $r = \log \frac{\pi_\theta}{\pi_{ref}}$ *can be used as PRM and ORM at the same time only if the following assumptions are satisfied: 1) The outcome reward is the total reward of the entire episode; 2)* $V(s_t) - V(s_{t-1}) = r(s_t, a_t)$.

These are strong assumptions especially the first one. For example, if the given outcome reward annotation is just whether the final output is correct or not, this method actually just distribute this outcome reward to each step without considering any other intermediate reward.

Furthermore, the reward function objective of PRIME is a cross-entropy loss, which is equivalent to a simplified version of our method without the importance sampling weight, $\max E_{\tau \sim \pi_{\text{ref}}}[\exp(r_\phi(\tau))] - E_{\tau \sim \pi_\theta}[\exp(r_\phi(\tau))]$. For the policy training, PRIMR uses a SOTA policy gradient method, which is our policy objective without the entropy constraint. As such, PRIME is a simplified version of our method with extra assumptions.

In summary, all the discussed techniques can be integrated into our proposed framework by introducing additional assumptions. In contrast, our framework makes the fewest assumptions and is therefore the most general, but also the most complicated learning process. The proof of all theorem and propositions can be found in the Appendix B.

## 4 EVALUATION

### 4.1 EXPERIMENT SETUP

**Tasks and datasets.** We assess the reasoning capabilities of our proposed method on seven benchmarks spanning competition-level mathematics and programming. For mathematics, we use AIME-2024 (Li et al., 2024a), AMC (Li et al., 2024a), Math-500 (Hendrycks et al., 2021), Minerva-Math (Lewkowycz et al., 2022), and Olympiad-Bench (He et al., 2024); for coding, we use Leetcode (Guo et al., 2024) and LiveCodeBench (Jain et al., 2024). We evaluate all models, including baselines, using greedy decoding and report the zero-shot pass@1 accuracy—the percentage of problems solved correctly on the first attempt. Our training data is constructed by sampling 7,000 instances per domain (*i.e.,* math and coding) from the PRIME Eurus-2-RL-Data (Cui et al., 2025a) dataset. For each sampled problem, we generate four expert demonstration trajectories using Claude-3.7-sonnet (Anthropic, 2025), which serve as the expert data for inverse RL. The validation set uses all 2,050 available samples from PRIME Eurus-2-RL-Data validation set for model selection.

**Baseline.** We compare against six baselines. The first is behavioral cloning (BC) on the expert dataset, a standard benchmark for inverse RL. The second is KTO (Ethayarajh et al.), which learns from preference labels. The other three are representative process-reward model training methods: PRIME, which derives implicit rewards via a DPO-like loss; MCTS, which estimates intermediate rewards at each step by rolling out the expert policy multiple times; and RL Tango (Zha et al., 2025), which trains an auxiliary LLM to predict step-wise scores. The final baseline is the vanilla RLOO algorithm with outcome-only rewards.

**Model and training setup.** We selected *Qwen2.5-3B-Instruct* (Yang et al., 2024) and *Qwen3-4B-Base* (Yang et al., 2025) as our base models, chosen for their strong instruction-following and reasoning abilities, as well as their training efficiency. Our training process consists of two sequential stages. *First*, we fine-tune the base model on mathematical datasets and evaluate its performance on math benchmarks. *Second*, this math-specialized model is used to initialize training on coding datasets, followed by a final evaluation on coding benchmarks. A separate PRM is also initialized from the same base model. All experiments were conducted using VeRL (Sheng et al., 2025) on eight L40S GPU.

We train both the policy and reward models using AdamW (Loshchilov & Hutter, 2017) with a microbatch size of 1 and minibatch size of 32. The policy model uses a constant learning rate of $5 \times 10^{-7}$ with batch size 128, while the PRM uses $3 \times 10^{-8}$, also with batch size 128. Policy training

Table 2: **Performance comparison with prior methods on mathematical and coding benchmarks.** Our method achieves state-of-the-art performance among 3B- and 4B-scale models across both domains. All RL models are trained for two epochs, and the best-performing model is selected based on validation results.[1]

| Model | Mathematical Reasoning | | | | | | Coding Reasoning | | |
|---|---|---|---|---|---|---|---|---|---|
| | MATH500 | AIME2024 | MinervaMath | AMC | OlympiadBench | Avg. | Leetcode | LiveCodeBench | Avg. |
| Qwen2.5-3B-Instruct | 46.0 | 10.0 | 22.4 | 34.9 | 28.9 | 28.4 | 26.7 | 18.3 | 22.5 |
| BC | 52.0 | 3.3 | 8.8 | 22.8 | 15.9 | 20.6 | 28.3 | 14.5 | 21.4 |
| KTO | 54.5 | 4.8 | 18.5 | 29.5 | 24.2 | 26.3 | 28.9 | 16.8 | 22.9 |
| RL-Tango | 57.1 | 6.6 | 25.2 | 32.5 | 29.5 | 30.2 | 30.8 | 20.3 | 25.6 |
| MCTS | 58.0 | 6.6 | 24.3 | 33.7 | 30.7 | 30.7 | 31.1 | 20.2 | 25.7 |
| PRIME | 56.2 | 6.6 | 26.5 | 31.3 | 29.0 | 29.9 | 30.0 | 20.4 | 25.2 |
| RLOO | 63.6 | 3.3 | 26.1 | 36.1 | 29.6 | 31.7 | 28.9 | 19.4 | 24.1 |
| **rePIRL** | 62.4 | 10.0 | 27.2 | 38.5 | 29.3 | **33.5** | 35.0 | 20.3 | **27.7** |
| Qwen3-4B-Base | 58.6 | 6.0 | 19.9 | 36.1 | 32.0 | 30.5 | 41.1 | 27.0 | 34.0 |
| BC | 60.0 | 4.0 | 14.5 | 28.0 | 22.0 | 25.7 | 38.5 | 24.5 | 31.5 |
| KTO | 66.0 | 5.5 | 24.0 | 36.5 | 34.5 | 33.3 | 42.5 | 26.5 | 34.5 |
| RL-Tango | 71.5 | 8.0 | 30.5 | 40.5 | 38.5 | 37.8 | 45.0 | 23.5 | 34.3 |
| MCTS | 70.6 | 6.0 | 29.0 | 37.3 | 37.6 | 36.1 | 45.5 | 27.4 | 36.4 |
| PRIME | 72.4 | 10.0 | 32.4 | 43.3 | 38.4 | 39.3 | 43.3 | 19.0 | 31.1 |
| RLOO | 73.0 | 10.0 | 29.8 | 39.7 | 39.7 | 38.4 | 46.1 | 24.9 | 35.5 |
| **rePIRL** | 72.8 | 16.0 | 33.5 | 43.3 | 42.5 | **41.6** | 52.2 | 27.5 | **39.8** |

follows RLOO with 4 rollouts per prompt, and the final reward is a weighted combination of outcome and PRM rewards, using a ratio of $1:0.05$ for *Qwen2.5-3B-Instruct* and $1:0.1$ for *Qwen3-4B-Base*. We also include a policy entropy loss with a coefficient 0.001. All models are trained for 2 epochs, and the best checkpoint is selected based on validation performance. For baseline models, we adopt default hyper-parameters whenever available; otherwise, we match those of our method to ensure fairness. Specifically, for the MCTS baseline, we use the PRM from Math-Shepherd (Wang et al., 2023). Additional training details are provided in the Appendix C.

## 4.2 MAIN EXPERIMENTS

We compare our method against six baselines on seven mathematical and coding benchmarks. Table 2 shows that our approach achieves state-of-the-art performance among 3B- and 4B-scale reasoning LLMs, reaching 33.5% (math) and 27.7% (coding) with *Qwen2.5-3B-Instruct*, and 41.6% (math) and 39.8% (coding) with *Qwen3-4B-Base*. Offline methods (*i.e.,* BC or KTO) underperform compared to RL-based approaches, highlighting the importance of online policy updates. For the PRM-based baselines, while they improve over the base Qwen model, they still lag behind our approach (*e.g.,* on the math dataset, we have about 2–3% higher accuracy than either of these PRM-based methods across all base model), further validating the effectiveness of our IRL-generated process reward.

Finally, compared to the vanilla RLOO algorithm, our method achieves absolute improvements of 2.8% in math and 3.6% in coding with *Qwen2.5-3B-Instruct*, and 3.2% in math and 4.3% in coding with *Qwen3-4B-Base*, demonstrating that our process rewards provide informative guidance beyond the solely final outcome signal.

## 4.3 ABLATION STUDY

Unless otherwise specified, all ablation study experiments are conducted using the *Qwen2.5-3B-Instruct* model.

**Choice of the policy update method.** We evaluate rePIRL with different RL algorithms by modifying only the advantage estimation while keeping the clipped surrogate loss fixed. We implement rePIRL with PPO, GRPO, and RLOO, where PPO learns an additional value function and GRPO normalizes expected total reward. As shown in Table 3, rePIRL consistently improves performance across all algorithms, demonstrating broad applicability. RLOO with process rewards achieves the best overall performance, while PPO shows minimal improvement despite added computational cost from its

---

[1] To ensure deterministic outputs across different evaluation runs, we follow the recommendation in vLLM Contributors (2025).

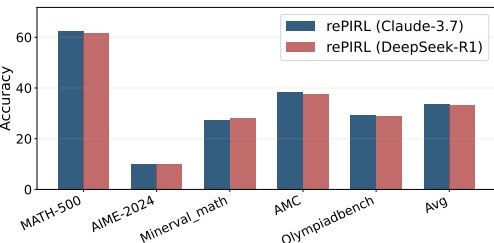
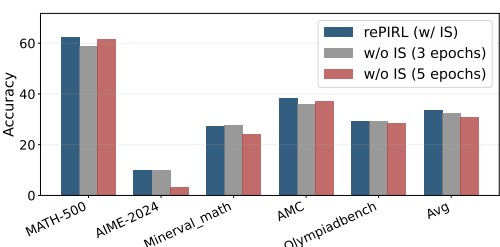

Figure 1: Comparison of rePIRL using Claude-3.7-Sonnet versus DeepSeek-R1 as expert trajectory generators.

Figure 2: Ablation study comparing rePIRL with importance sampling versus baselines with only increased policy updates.

critic model. These results indicate RLOO provides the most effective advantage estimation for process-based rewards, leading us to adopt it as the default method in subsequent experiments.

Table 3: **Performance comparison of different RL algorithms on mathematical and coding benchmarks**. Across all tasks, RLOO with our PRM achieves the best average performance.

| | Mathematical Reasoning | | | | | | Coding Reasoning | | |
|---|---|---|---|---|---|---|---|---|---|
| Model | MATH500 | AIME2024 | MinervaMath | AMC | OlympiadBench | Avg. | Leetcode | LiveCodeBench | Avg. |
| Qwen2.5-3B-Instruct | 46.0 | 10.0 | 22.4 | 34.9 | 28.9 | 28.4 | 26.7 | 18.3 | 22.5 |
| PPO | 51.4 | 3.3 | 25.0 | 31.3 | 28.3 | 27.9 | 27.2 | 18.9 | 23.0 |
| PPO w/ rePIRL | 56.6 | 13.3 | 25.0 | 28.9 | 27.9 | 30.3 | 33.5 | 20.1 | 26.8 |
| GRPO | 61.6 | 3.3 | 26.1 | 36.1 | 28.6 | 31.1 | 29.0 | 19.2 | 24.1 |
| GRPO w/ rePIRL | 62.2 | 10.0 | 27.4 | 36.5 | 29.1 | 33.0 | 34.3 | 20.1 | 27.2 |
| RLOO | 60.1 | 3.3 | 25.7 | 35.2 | 29.1 | 31.7 | 28.9 | 19.4 | 24.1 |
| RLOO w/ rePIRL | 62.4 | 10.0 | 27.2 | 38.5 | 29.3 | **33.5** | 35.0 | 20.3 | **27.7** |

**Testing-time scaling.** To improve performance at test time, we generate multiple candidate solutions and select an outcome reward model (Liu et al., 2024) for answer selection. We group solutions by their final answers and compute the total reward score for each group (Li et al., 2023a). The final prediction is selected from the highest-scoring group, choosing the solution with the maximum individual reward score within that group. The results are shown in Figure 3a, rePIRL demonstrates superior test-time scaling behavior on AIME2024, achieving significant improvements from 0.1 to 0.2 accuracy as the number of rollouts increases from 1 to 16.

**Replacing Claude with open-source models.** We further evaluate rePIRL by replacing the Claude-3.7-Sonnet model (already not the latest Claude model) with the open-source DeepSeek-R1-Reasoning model (Guo et al., 2025) for collecting expert trajectories. The performance on math tasks is shown in Figure 1. The comparable results using Claude and Deepseekp-R1-Reasoning as expert models further demonstrate that rePIRL does not rely on costly proprietary models to generate expert trajectories.

**Importance sampling vs update epochs.** Note that we apply importance weighting when updating the reward model in Section 3.2. We additionally compare our method against a baseline that simply increases the number of inner policy updates (*i.e.,* motivated by standard GAN training). Across all math tasks, our method consistently outperforms this heuristic baseline as shown in Figure 2. These results demonstrate that the performance gains stem from the necessity of the importance-sampling correction itself, rather than from merely performing additional policy updates.

We also conduct additional experiments to evaluate the generalizability of our reward model across different architectures. Details are provided in Appendix D.1.

## 4.4 TRAINING WITH THE PRM ONLY

To isolate the effect of our dense, process-based reward signal, we conduct an ablation study in which the RL agent is trained solely on intermediate rewards from the PRM. In the main experiments, the reward function combines step-wise PRM scores with a binary outcome reward; here, we remove the outcome component entirely. As a result, the policy is optimized only to generate reasoning trajectories that the PRM rates highly, without receiving any direct feedback on final correctness. This setup enables us to assess how effectively process-based rewards alone can guide the model toward correct solutions.

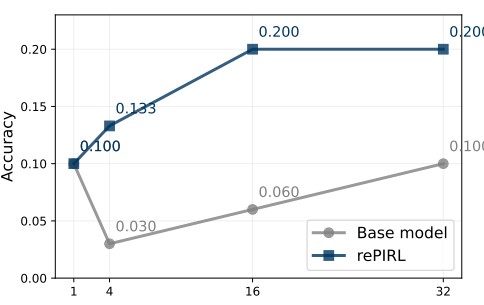 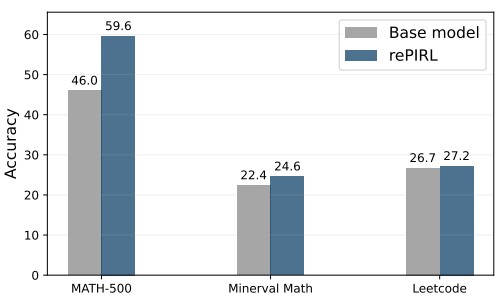

(a) Test time scaling on AIME2024.  (b) Performance comparison on three datasets.

Figure 3: Comparison of base model and reIRL performance across different settings.

We evaluate this variant on Math-500, MinervaMath, and Leetcode based on *Qwen2.5-3B-Instruct*. The results in Figure 3b show that training exclusively with process-level rewards yields substantial gains over the Qwen base model, including a $13.6\%$ improvement on Math-500. These findings suggests that our IRL framework effectively recovers the latent reward function consistent with the expert policy.

## 5 DISCUSSION

**Training Efficiency.** We emphasize that rePIRL does not introduce significant computational overhead compared to many state-of-the-art methods. By leveraging RLOO for non-parametric value estimation, rePIRL eliminates the need for a value network, restricting optimization to only the policy and reward models. This architectural complexity aligns with methods requiring PRMs, such as PRIME and Math-Shepherd. For instance, when fine-tuning *Qwen2.5-3B-Instruct* on math tasks, rePIRL requires approximately 13.3 training hours compared to 14.6 hours for PRIME. Beyond efficiency, rePIRL provides a unified framework; as detailed in Section 3.3. Empirically, this formulation results in consistent performance gains over all baselines across diverse tasks and model architectures.

**Training Stability.** Adversarial training is well known to suffer from instability, a pattern we also observed in our preliminary experiments. To mitigate this, we enhance reward model training with corrected trajectories from the current policy, treating them as additional expert data while retaining only failed trajectories as policy data (Algorithm 1). This signals to the reward model that these corrected trajectories should receive high reward, yielding more reliable gradients for policy updates. As shown in our ablation study (Appendix D.2), this component substantially improves both training stability and final performance.

## 6 CONCLUSION AND FUTURE WORKS

In this work, we presented rePIRL, an IRL–inspired framework for learning PRMs with minimal assumptions. We introduce a dual learning process that alternates between policy and PRM updates, along with techniques tailored to scaling inverse RL for LLMs. We prove that rePIRL can unify online and offline PRM learning under additional assumptions, and our experiments on math and coding reasoning benchmarks demonstrate clear improvements. Ablation studies further validated the importance of our key design choices.

First, it would be valuable to systematically study and compare the various heuristics proposed for online RL with outcome reward for LLM reasoning (*e.g.,* curriculum learning (Bengio et al., 2009), self-correction (Shinn et al., 2023)), comparing their effectiveness under our learning framework. Second, rePIRL could be combined with complementary approaches for learning PRMs, *e.g.,* the intrinsic reward methods. It is interesting to investigate the proper reward shaping strategies when comparing rePIRL with other PRMs as well as different outcome rewards. Finally, investigating the effectiveness of scaling rePIRL to agentic tasks and larger models may unlock the potential of PRMs for broader application domains and enable stronger performance.

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

## A CLARIFICATION ON THE USE OF LLM

The authors acknowledge the use of a large language model, employed exclusively to assist with grammar correction, proofreading, and minor stylistic improvements throughout the manuscript. The role of LLMs is strictly limited to language polishing and does not rise to the level of a contributor to the research. All core research, methodology, and substantive claims are the original work of the authors, who remain solely responsible for the content and conclusions presented.

## B THEORETICAL PROOF

### B.1 PROOF OF THEOREM 1

We consider the entropy-regularized objective at a fixed state $s$ with action-values $Q^*(s, a)$:

$$\max_{\pi(\cdot|s)} \mathcal{L}(\pi) = \sum_a \pi(a|s)Q^*(s, a) + \beta\Big[-\sum_a \pi(a|s)\log \pi(a|s)\Big], \qquad \sum_a \pi(a|s) = 1.$$

Introducing a Lagrange multiplier $\lambda$ for the normalization constraint, the Lagrangian is

$$\mathcal{J}(\pi, \lambda) = \sum_a \pi(a)\Big[Q^*(s, a) - \beta \log \pi(a)\Big] + \lambda\Big(1 - \sum_a \pi(a)\Big).$$

Setting the derivative w.r.t. $\pi(a)$ to zero:

$$\frac{\partial \mathcal{J}}{\partial \pi(a)} = Q^*(s, a) - \beta(1 + \log \pi(a)) - \lambda = 0 \quad \Rightarrow \quad \pi(a) \propto \exp\Big(\frac{Q^*(s, a)}{\beta}\Big).$$

Normalizing over $a$ yields the optimal policy:

$$\pi^*(a|s) = \frac{\exp\big(Q^*(s, a)/\beta\big)}{\sum_{a'} \exp\big(Q^*(s, a')/\beta\big)}.$$

Substituting $\pi^*$ into $V^*$ gives the soft value function

$$V^*(s) = \beta \log \sum_{a'} \exp\big(Q^*(s, a')/\beta\big),$$

so that $\pi_*(a|s) = \exp(\frac{1}{\beta}[Q^*(s, a) - V^*(s)])$.

### B.2 PROOF OF PROPOSITION 1

Starting from the entropy-regularized optimal policy: $\pi_*(a|s) = \exp(\frac{1}{\beta}[Q^*(s, a) - V^*(s)])$

Taking logs and subtracting $\log \pi_{\text{ref}}(a|s)$:

$$\log \frac{\pi^*(a|s)}{\pi_{\text{ref}}(a|s)} = \frac{Q^*(s, a) - V^*(s)}{\beta} - \log \pi_{\text{ref}}(a|s).$$

In the bandit case, $Q^*(s,a) = r'(s,a) = r(s,a) + \beta \log \pi_{\text{ref}}(a|s)$, giving

$$\log \frac{\pi^*(a|s)}{\pi_{\text{ref}}(a|s)} = \frac{r(s,a) - V^*(s)}{\beta}.$$

Exponentiating both sides leads to

$$\pi_*(a \mid s) = \frac{1}{Z_*(s)} \pi_{\text{ref}}(a \mid s) \exp\left(\frac{1}{\beta} r(a,s)\right), \quad Z_*(s) = \exp\left(\frac{V^*(s)}{\beta}\right).$$

### B.3    PROOF OF PROPOSITION 2

In DQO, the authors adopt the Soft Actor-Critic (SAC) framework to learn the state-value function $V$ and state-action value function $Q$. In SAC, these functions are optimized by minimizing the squared Bellman residuals:

$$L_V(\phi) = \mathbb{E}_{(s_t,a_t,s_{t+1})\sim\mathcal{D}}\left[\left(V_\phi(s_t) - Q_\theta(s_t,a_t) + \beta \log \pi_\theta(a_t \mid s_t)\right)^2\right], \quad \text{(b.3.1)}$$

$$L_Q(\theta) = \mathbb{E}_{(s_t,a_t,s_{t+1})\sim\mathcal{D}}\left[\left(Q_\theta(s_t,a_t) - r(s_t,a_t) - V_\phi(s_{t+1})\right)^2\right]. \quad \text{(b.3.2)}$$

At convergence, these updates satisfy the soft Bellman equations:

$$\hat{V}^\pi(s_t) = \mathbb{E}_{a_t\sim\pi(\cdot|s_t)}\left[\hat{Q}^\pi(s_t,a_t) - \beta \log \pi(a_t \mid s_t)\right], \quad \hat{Q}^\pi(s_t,a_t) = r(s_t,a_t) + \gamma\,\mathbb{E}_{s_{t+1}}\left[\hat{V}^\pi(s_{t+1})\right].$$

which correspond to the soft maximum-entropy RL solution in Theorem 1.

Motivated by DPO, DQO further reparameterizes the $Q$-function directly in terms of the policy:

$$Q_\theta(s_t,a_t) = V_\phi(s_t) + \beta \log \pi_\theta(a_t \mid s_t), \quad \text{(b.3.3)}$$

where $\pi_\theta$ denotes the policy network. Notably, (b.3.3) coincides with the optimal policy $\pi^*$ under the soft maximum-entropy RL formulation, as established in Theorem 1. By substituting (b.3.3) into (b.3.2) and rewriting (b.3.1) by replacing the $Q$-function using (b.3.2) accordingly, DQO eliminates the explicit dependence on the $Q$-function. Taken together, DQO optimizes the same maximum-entropy RL objective for the policy model as our method.

### B.4    PROOF OF PROPOSITION 3

Our main idea is to show that the gradient of Eqn. (5) is equivalent to the gradient of $\mathbb{E}_{\tau\sim\mathcal{D}}\left[\log D(\tau)\right] + \mathbb{E}_{\tau\sim\pi_\theta}\left[\log\left(1 - D(\tau)\right)\right]$, where $D = \frac{\tilde{p}}{\tilde{p}+\pi_\theta}$ and $\tilde{p} = \frac{1}{z}\exp(r_\phi(\cdot))$, thereby establishing the equivalence between the two objective functions.

To begin with, the gradient of Eqn. (5) can be written as

$$\partial_\phi \mathcal{L}_{\text{reward}}(\phi) = \mathbb{E}_{\tau\sim\mathcal{D}}[\partial_\phi r_\phi(\tau)] - \partial_\phi \log\left(\mathbb{E}_{\tau\sim\mu}\left[\frac{\exp(r_\phi(\tau))}{\tilde{\mu}(\tau)}\right]\right)$$

$$= \mathbb{E}_{\tau\sim\mathcal{D}}[\partial_\phi r_\phi(\tau)] - \frac{\mathbb{E}_{\tau\sim\mu}\left[\frac{\exp(r_\phi(\tau))}{\tilde{\mu}(\tau)}\partial_\phi r_\phi(\tau)\right]}{\mathbb{E}_{\tau\sim\mu}\left[\frac{\exp(r_\phi(\tau))}{\tilde{\mu}(\tau)}\right]}$$

$$= \mathbb{E}_{\tau\sim\mathcal{D}}[\partial_\phi r_\phi(\tau)] - \mathbb{E}_{\tau\sim\mu}\left[\frac{\frac{1}{z}\exp(r_\phi(\tau))\partial_\phi r_\phi(\tau)}{\tilde{\mu}(\tau)}\right].$$

Here, $\mu$ denotes a mixture distribution of $\pi_\theta$ and $D$, where $\tilde{\mu}(\tau) = \frac{1}{2Z}\exp\left(r_\phi(\tau)\right) + \frac{1}{2}\pi_\theta(\tau)$.

Note that, $\mathbb{E}_{\tau \sim \mathcal{D}}\big[\log D(\tau)\big] + \mathbb{E}_{\tau \sim \pi_\theta}\big[\log\big(1 - D(\tau)\big)\big]$ with $D = \frac{\tilde{p}}{\tilde{p} + \pi_\theta}$ and $\tilde{p} = \frac{1}{z}\exp(r_\phi(\cdot))$ can be written as

$$\mathcal{L}_{\text{entropy}}(D_\phi) = \mathbb{E}_{\tau \sim \mathcal{D}}\big[\log D(\tau)\big] + \mathbb{E}_{\tau \sim \pi_\theta}\big[\log(1 - D(\tau))\big]$$

$$= \mathbb{E}_{\tau \sim \mathcal{D}}\left[\log \frac{\frac{1}{z}\exp(r_\phi(\tau))}{\frac{1}{z}\exp(r_\phi(\tau)) + \pi_\theta(\tau)}\right] + \mathbb{E}_{\tau \sim \pi_\theta}\left[\log \frac{\pi_\theta(\tau)}{\frac{1}{z}\exp(r_\phi(\tau)) + \pi_\theta(\tau)}\right]$$

$$= \mathbb{E}_{\tau \sim \mathcal{D}}\left[\log \frac{\frac{1}{z}\exp(r_\phi(\tau))}{\tilde{\mu}(\tau)}\right] + \mathbb{E}_{\tau \sim \pi_\theta}\left[\log \frac{\pi_\theta(\tau)}{\tilde{\mu}(\tau)}\right]$$

$$= -\log z + \mathbb{E}_{\tau \sim \mathcal{D}}[r_\phi(\tau)] - \mathbb{E}_{\tau \sim \mathcal{D}}[\log \tilde{\mu}(\tau)] + \mathbb{E}_{\tau \sim \pi_\theta}[\log \pi_\theta(\tau)] - \mathbb{E}_{\tau \sim \pi_\theta}[\log \tilde{\mu}(\tau)]$$

$$= -\log z + \mathbb{E}_{\tau \sim \mathcal{D}}[r_\phi(\tau)] + \mathbb{E}_{\tau \sim \pi_\theta}[\log \pi_\theta(\tau)] - 2\mathbb{E}_{\tau \sim \mu}[\log \tilde{\mu}(\tau)].$$

Differentiating these terms yields

$$\partial_\phi \mathcal{L}_{\text{entropy}}(D_\phi) = \mathbb{E}_{\tau \sim d}[\partial_\phi r_\phi(\tau)] - \mathbb{E}_{\tau \sim \mu}\left[\frac{\frac{1}{Z}\exp(r_\phi(\tau))\partial_\phi r_\phi(\tau)}{\tilde{\mu}(\tau)}\right] = \partial_\phi \mathcal{L}_{\text{reward}}(\phi).$$

which completes the proof.

### B.5 Proof of Proposition 4

We begin with the original proof that derives the intermediate reward in PRIME and outline the assumptions underlying its formulation.

Specifically, PRIME assumes that the final reward can be expressed as $r = \log \frac{\pi_\theta}{\pi_{\pi_{\text{ref}}}}$ and derives the intermediate reward $r_t$ from the difference in the value function between consecutive time steps:

$$\sum_{i=1}^{t} \beta \log \frac{\pi_\theta(y_i|\mathbf{y}_{<i})}{\pi_{\pi_{\text{ref}}}(y_i|\mathbf{y}_{<i})} - \sum_{i=1}^{t-1} \beta \log \frac{\pi_\theta(y_i|\mathbf{y}_{<i})}{\pi_{\pi_{\text{ref}}}(y_i|\mathbf{y}_{<i})}.$$

Taken together, the ORM is defined over the entire sequence (the outcome), indicating that the outcome reward corresponds to the total reward of the episode. This final reward is perfectly decomposable and can be expressed as a sum of per-step rewards, as shown by $V(s_t) - V(s_{t-1}) = r(s_t, a_t)$.

## C Experiment Details

### C.1 Training and Inference Details

**Training Details.** For training both our methods and the baselines, we set the maximum prompt length to 1,535 tokens and the maximum response length to 3,000 tokens. Following PRIME, we filter out prompts for which the accuracy exceeds 0.8 or falls below 0.2. For the clipping hyperparameter, we use the default clip ratio of 0.2 for both lower and upper bounds. The policy model uses a gradient clip value of 1.0, while the reward model uses a value of 10.0. The complete set of training hyper-parameters is detailed in Table 4.

**Inference Parameters.** For inference, we adopt a greedy decoding strategy (*i.e.,* setting the temperature to 0, `top_k` to 1, and `top_p` to 1) for all models. Specifically, we perform inference using the latest version of the VLLM (Kwon et al., 2023) framework to reduce memory usage and accelerate computation. For the testing-time scaling experiments in Section 4.3, we set the temperature to 0.8 while keeping all other hyper-parameters unchanged.

## D Additional Experiments

### D.1 Reward Model Architecture Ablation

We conducted additional ablation studies on the reward model's architecture. The current rePIRL results utilize a PRM initialized from the same base language model as the policy model. We systematically varied the reward model sizes and architectures as follows.

| config | value |
|---|---|
| optimizer | AdamW |
| policy model learning rate | 5e-7 |
| reward model learning rate | 3e-8 |
| weight decay | 0.0 |
| policy model batch size | 128 |
| reward model batch size | 128 |
| policy model mini batch size | 32 |
| reward model mini batch size | 32 |
| policy model micro batch size | 1 |
| reward model micro batch size | 1 |
| max prompt length | 1535 |
| max response length | 3000 |
| reward model gradient clip value | 10.0 |
| policy model gradient clip value | 1.0 |
| clip ratio | 0.2 |
| policy model epoch | 1 |
| reward model epoch | 1 |
| total training epochs | 2 |
| coefficient of entropy policy loss | 0.001 |

Table 4: Training hyperparameters for both coding and math dataset.

Table 5: Performance comparison across different reward model configurations on math tasks. Here, P-3B-R-1.5B denotes that the policy model is *Qwen2.5-3B-Instruct* and reward model is *Qwen2.5-1.5B-Instruct*.

| Dataset | MATH-500 | AIME-2024 | Minerval math | AMC | Olympiadbench | Avg |
|---|---|---|---|---|---|---|
| P-3B-R-3B | 62.4 | 10.0 | 27.2 | 38.5 | 29.3 | 33.5 |
| P-3B-R-1.5B | 57.8 | 10.0 | 27.6 | 36.1 | 27.6 | 31.8 |
| P-4B-R-4B | 72.8 | 16.0 | 33.5 | 43.3 | 42.5 | 41.6 |
| P-4B-R-1.7B | 72.6 | 10.0 | 34.9 | 37.3 | 38.7 | 38.7 |

From Table 5, we observe that replacing the reward model with a smaller one degrades performance. Nevertheless, our approach still outperforms the RLOO baselines, demonstrating that rePIRL generalizes across different reward model architectures and sizes. We note that using Qwen models for experiments and ablation is standard practice, as the Qwen family represents the best-performing open-source model series. Different Qwen models have distinct architectural properties that enable controlled comparisons. We do not consider other LLM families (*e.g.*, LLaMA) or non-LLM architectures, given the performance differences in their base models.

## D.2 ROLLOUT STRATEGY ABLATION

Table 6: Performance comparison across different rollout strategies on math tasks. Here, rePIRL w/o correct traj from policy indicates that correct trajectories produced by the policy are not included as pseudo-expert trajectories. We utilize the *Qwen2.5-3B-Instruct* as our base model.

| Dataset | MATH-500 | AIME-2024 | Minerval math | AMC | Olympiadbench | Avg |
|---|---|---|---|---|---|---|
| rePIRL | 62.4 | 10.0 | 27.2 | 38.5 | 29.3 | 33.5 |
| rePIRL w/o correct traj from policy | 61.4 | 3.3 | 31.6 | 34.9 | 27.6 | 31.7 |

