# OpenReview forum: "rePIRL: Learn PRM with Inverse RL for LLM Reasoning"
_ICLR.cc/2026/Conference — Submitted to ICLR 2026_

### Official Review · Reviewer_TGya · 2025-10-20

**Soundness:** 1
**Presentation:** 2
**Contribution:** 1
**Rating:** 2
**Confidence:** 4

**Summary:**

This paper proposes to learn a process reward from expert demonstrations via inverse reinforcement learning. Specifically, the paper parameterizes the trajectory distribution using energy model and solves a maximum likelihood estimation problem to learn the process reward. The algorithm alternates between reward update and policy update, which is a standard practice of IRL. The experiment uses Qwen-2.5-3B to demonstrate the effectiveness of the proposed approach.

**Strengths:**

The paper aims to address the issue of absence of process reward, which is an important problem for RL to train language models.

**Weaknesses:**

1. The idea of using IRL to learn a reward function from expert demonstrations is not new, please see the reference [1,2]. What is your difference?

2. The theoretical part of this paper is simple copy-paste from literature without mentioning them. For example, the claim in Theorem 1 is just soft policy. This result and also proof have been provided in literature, with continuous version provided in [3] and discrete version provided in [4]. You proof in Appendix B.1 has the same logic and is a simplification of the proof in [4]. The propositions are just simple extension of Theorem 1. I highly suggest the authors explicitly mention the reference, revise Section 3.3, and point out that the theoretical results come from the reference [3,4], otherwise, it is misleading.

3. I find the definition of $p(o_t|s_t,a_t)$ in line 197 problematic. This energy model only depends on the current reward $r_\phi(s_t,a_t)$ which does not make sense. In RL, the optimal action is the action that has the highest Q-value, i.e., that can maximize the long-term return (i.e., cumulative reward). That says, the energy model should depend on the Q-value. If it only depends on the current reward, it means that the action is myopic, i.e., it only wants to maximize the instant reward instead of the long-term return. This kind of action, in general, cannot be the optimal action, unless your reward function is a Q-function which is difficult to learn/define in practice.

[1] Getting More Juice Out of the SFT Data: Reward Learning from Human Demonstration Improves SFT for LLM Alignment

[2] From Demonstrations to Rewards: Alignment Without Explicit Human Preferences

[3] Soft Actor-Critic: Off-Policy Maximum Entropy Deep Reinforcement Learning with a Stochastic Actor

[4] Infinite Time Horizon Maximum Causal Entropy Inverse Reinforcement Learning (the TAC version)

**Questions:**

Please see weaknesses.

---

> ### Author Response · Authors · 2025-11-24
> **Rebuttal (part-1)**
>
> >W1: Comparison with two other works [1,2] that learns reward from expert demonstrations.
>
> A1: Thanks for raising these two works. We would like to kindly point out that these two methods are fundamentally different from rePIRL in the following aspects. (1) These two methods are designed for alignment rather than reasoning. As such, the fundamental model is different. As stated in their papers, alignment mainly optimizes over the entire trajectories $(x, y)$, which fundamentally follows a bandit assumption (as discussed in our Section 3). However, reasoning requires modeling the problem as a real MDP, making the problem much more challenging. (2) To solve the challenge, we developed rePIRL under a token-level MDP and addressed the scalability (sample from unknown optimal policy) and intractability (partition function $Z$) issues with a series of new algorithmic designs. Our technical designs enable reliable recovery of an effective Process Reward Model (PRM) that provides token-level reward during LLM reasoning, leading to policies with stronger performance. (3) rePIRL conducted a rigorous theoretical analysis of major existing post-training methods and integrated them into the same framework with different assumptions. This analysis is helpful for obtaining a holistic understanding of RL-driven LLM post-training. As a result, rePIRL is the first work that generalizes classical IRL for LLM reasoning via a scalable learning algorithm. We believe this is a non-trivial contribution as there are many other papers accepted at top-tier ML conferences that generalize classical RL theorems or frameworks to scalable deep learning settings [1,2,3]. Will clarify the difference in the revised paper.
>
> >W2: Limited contribution of the theoretical part.
>
> A2: Thanks for pointing out our reliance on maximum-entropy reinforcement learning (MaxEnt RL). We agree with the review on this and will clarify the related works on Theorem 1 to clarify confusion. Yes, it is a well-established result. It has been used as the foundational theorem for many papers accepted at top-tier ML conferences [4,5]. It is a necessary foundation for us as well.
> However, we would like to respectfully point out that relying on theorem 1 does not dilute our theoretical contribution. Our key theoretical contributions are as follows. **We proposed a unified post-training framework that includes SOTA reasoning methods, especially DPO, DQO, PRIME, and MCTS**. For each method, we analyzed its underlying assumptions and connections to our framework (Propositions 1–4). For example, DPO emerges as a special case requiring a specific reward function form and the bandit assumption (Proposition 1). PRIME assumes that the outcome reward equals the total trajectory reward and that the value-function difference perfectly reflects the immediate reward (Proposition 4). Such an analysis requires non-trival effort and is all novel and original results. Through this analysis, we establish a unified framework, a better understanding of different reasoning methods, and a conclusion that rePIRL is a generalized PRM learning method that operates under the minimal necessary assumptions. This supports the superior empirical performance in our experiment.
> We apologize for the confusion again. We will make it clear in the revision that Theorem 1 is our foundation rather than novelty. Using this foundation, we formally prove that several SOTA PRM learning algorithms are restricted instances derived from our broader Inverse RL framework. Will clarify this in the revision.

---

> ### Author Response · Authors · 2025-11-24
> **Rebuttal (part-2)**
>
> >W3: I find the definition of $p(o_t|s_t, a_t)$ in line 197 problematic. This energy model only depends on the current reward  which does not make sense. In RL, the optimal action is the action that has the highest Q-value, i.e., that can maximize the long-term return (i.e., cumulative reward). That says, the energy model should depend on the Q-value. If it only depends on the current reward, it means that the action is myopic, i.e., it only wants to maximize the instant reward instead of the long-term return. This kind of action, in general, cannot be the optimal action, unless your reward function is a Q-function which is difficult to learn/define in practice.
>
>
> A3: Thanks for the comment. We would like to first respectfully clarify that $p(o_t|s_t, a_t)$ is intentionally defined as the probability of the current action being sampled for the expert policy, where $o_t$ is a hidden indicator, equaling either 0 or 1. As such, it is designed to depend only on the current state and action. It is **a correct definition**. Second, we would like to further emphasize that this definition does not affect our fundamental mechanism of RL that relies on the entire trajectories. This is first supported by our token-level MDP model, which naturally defines the value function as the long-term accumulated reward, which is always our learning objective. For example, as stated in line 198 of our paper, "$p(\tau|o\_{1:T})$ is the probability of trajectory $\tau$ being sampled from the expert policy."
>
> $$p(\tau|o\_{1:T}) \propto p(\tau)\exp\left(\sum\_t r\_\phi(s\_t, a\_t)\right)$$
>
> This formulation ensures that the reward model considers the total return accumulated across the full trajectory rather than a single-step reward. Furthermore, the reward model is trained via a maximum-likelihood inverse RL objective that explicitly optimizes for the **total accumulated reward** over expert trajectories:
>
> $$
> \max\_{\phi} \mathbb{E}\_{\tau \sim \mathcal{D}}[r\_\phi(\tau)] - \log \left( \mathbb{E}\_{\bar{\tau} \sim \pi\_\theta} \left[ \frac{\exp(r\_\phi(\bar{\tau}))}{\pi\_\theta(\bar{\tau})} \right] \right)
> $$
>
> where $r\_\phi(\bar{\tau}) = \sum\_t r\_\phi(s\_t, a\_t)$ represents the **cumulative reward** over the trajectory. The associated importance weight used in gradient estimation is:
>
> $$
> w\_j = \frac{\exp\left(\sum\_t r\_\phi(s\_t, a\_t)\right)}{\prod\_t \pi\_\theta(a\_t|s\_t)} = \frac{\exp(r\_\phi(\bar{\tau}\_j))}{\pi\_\theta(\bar{\tau}\_j)}
> $$
>
> The exponential term $\exp(r\_\phi(\bar{\tau}\_j))$ thus captures the **trajectory-level cumulative reward**, not the immediate or myopic reward. Mathematically, this confirms that the reward model and corresponding energy function in our framework align with long-term cumulative return—consistent with the principles of optimal policy learning in RL.
>
> References:
>
> [1] Learning Robust Rewards with Adversarial Inverse Reinforcement Learning, ICLR 2018
>
> [2]  Q♯: Provably Optimal Distributional RL for LLM Post-Training, Neurips 2025
>
> [3] PC-PG: Policy Cover Directed Exploration for Provable Policy Gradient Learning, Neurips 2020
>
> [4] Reinforcement Learning with Deep Energy-Based Policies, ICML 2017
>
> [5] From r to Q∗ : Your Language Model is Secretly a Q-Function, COLM 2024

---

> > ### Author Response · Authors · 2025-11-26
> >
> > Dear Reviewer,
> >
> > We sincerely appreciate your thoughtful feedback. As we enter the final week of the discussion period, we would be grateful to know whether our recent responses have addressed your concerns or if any questions remain. We are happy to provide any further clarification as needed. Wishing you a Happy Thanksgiving.
> >
> > Thank you,
> >
> > The Authors

---

> > > ### Comment · Reviewer_TGya · 2025-11-27
> > >
> > > Thanks for the response.
> > >
> > > W1. The authors argue that this paper studies IRL for MDP while the references [1,2] study bandit setting. Can the authors elaborate the technical difficulty of extending bandit IRL to MDP IRL other than simply changing reward comparison to cumulative reward comparison. Specifically, the fundamental spirit of reward update in IRL is reward comparison. Suppose that the expert trajectory is $s_0^E,a_0^E,\cdots$ in MDP setting (or $s_0^E,a_0^E$ in bandit setting) and the learner trajectory is $s_0,a_0,\cdots$ (or $s_0,a_0$ in bandit setting). The reward update mechanism of IRL is $\nabla_\theta [r_\theta(s_0^E,a_0^E)-r_\theta(s_0,a_0)]$ for bandit and $\nabla_\theta [\sum_{t=0}^{\infty}r_\theta(s_t^E,a_t^E)-\sum_{t=0}^{\infty}r_\theta(s_t,a_t)]$ for MDP. You can see that there is no much difference.
> > >
> > > W2. It is fine that the paper is based on well-esablished theoretical results from literature, which is not my concern. My concern is that the paper has formal theoretical statements that copy paste from the literature, giving the feeling that the paper has certain theoretical contributions (but actually does not). For the revised version, I still have the same feeling that this paper wants to package itself as a theoretical paper but actually does not have novel theoretical contributions. The authors mention that the core theoretical contribution is a unified post-training framework . However, I do not see why it is called "theoretical" contribution cause I do not see any major proof in appendix on this matter. In that sense, I do not see why this paper forces itself to look like a theoretical paper. It is fine that a paper does not have any theoretical contributions at all. However, it is misleading and even inappropriate for a non-theoretical paper to pretend to be a theoretical paper.
> > >
> > > W3. I understand that the trajectory distribution and the maximum likelihood part uses cumulative reward. My concern is only about this definition $p(o_t|s_t,a_t)$. In the paper, it is said that $o_t$ indicates that whether $a_t$ is optimal or not. I am very confused how this can be possible? Whether $a_t$ is optimial or not should depend on cumulative reward $\sum_{t'\geq t}r(s_{t'},a_{t'})$, but in the current definition, it only depends on the current reward $r(s_t,a_t)$. How can you tell whether an action $a_t$ is optimal or not by only looking at the myopic reward $r(s_t,a_t)$?

---

> ### Author Response · Authors · 2025-11-27
> **Response to W1**
>
> We thank the reviewer for the prompt responses and the valuable additional comments.
>
> Response to W1
>
> We would like to respectfully point out that extending Bandit IRL to the MDP setting—particularly for LLM reasoning—presents fundamental technical challenges that go far beyond a simple summation of rewards. We distinguish rePIRL from the referenced works [1,2] in three key aspects:
> 1. **Fundamental Model Difference (Alignment vs. Reasoning)**: As noted in the reviewer's cited works [1,2], those methods are primarily designed for alignment, which typically optimizes over entire trajectories (responses) and fundamentally follows a Bandit assumption (as discussed in our Section 3). In the Bandit setting, the action does not influence the state distribution. As such, these works actually treat the entire trajectory as one step. There is no multiple-step dependency. Fundamentally, the core for learning under the bandit assumption is just balancing between exploration and exploitation. No need to consider long-term rewards. However, reasoning tasks require modeling the problem as a real MDP because the correctness of a reasoning step depends on the sequential history, and current tokens dictate future reasoning paths. Unlike the Bandit setting, MDP IRL must solve the **credit assignment problem**, ensuring that intermediate steps (tokens) receive appropriate reward signals rather than just the final output.
> 2. **Technical Challenge: Intractability and Scalability**: The reviewer suggests the difference is merely changing reward comparison to cumulative reward comparison. However, this shift introduces the partition function intractability.
> In a Bandit setting, normalizing the policy distribution is often trivial or tractable over a discrete action set.
> In an MDP (especially with LLMs), the partition function involves integrating over an exponential number of possible trajectories. The core technical contribution of rePIRL is addressing this **intractability** (calculating $Z_\theta$) and **scalability** (sampling from an unknown optimal policy) via new algorithmic designs. These designs enable the reliable recovery of an effective token-level PRM, which is not possible under the Bandit assumptions used in [1,2].
> 3. **A Unified Theoretical Framework**: rePIRL conducted a rigorous theoretical analysis of major existing post-training methods and integrated them into the same framework with different assumptions. This analysis is helpful for obtaining a holistic understanding of RL-driven LLM post-training.
> We would also like to point out that even under the alignment setting, extending a learning method (e.g., DPO) to a multi-step MDP setting is non-trivial. The contribution was also publishable in top-tier ML conferences (e.g., [1]).
>
> [1] From r to q: Your language model is secretly a q-function, COML’24

---

> ### Author Response · Authors · 2025-11-27
> **Response to W2**
>
> Response to W2
>
>
> We respectfully disagree with the assessment that the paper lacks a theoretical contribution or merely "packages" itself as theoretical. To the best of our knowledge, we are the first to provide a **unified framework** that grounds these baseline methods in rigorous theoretical derivation, especially with the following conclusions that are worth highlighting.
>
> **Non-trivial Unification (Propositions 3 & 4)**: We rigorously demonstrate that methods like MathShepherd and PRIME are actually instantiations of our framework under specific conditions. For instance, in **Proposition 4** (and detailed in Appendix B), we prove that PRIME is a special case of our method that relies on strong assumptions—assumptions we show may not hold in complex real-world reasoning tasks. This is a novel theoretical insight, not a restatement.
>
> **Objective Alignment**: We theoretically demonstrate that our reward objective (Eq. 5) aligns with the cross-entropy loss used to train Process Reward Models (PRMs) in MathShepherd.
>
> We believe that identifying the common theoretical backbone of these empirical successes is a notable contribution to the community. In addition, we would like to respectfully point out that stating our theoretical analysis is simply "copy and paste" is not a fair argument.  Certain model settings and theorems of our paper are necessary for our framework. As such, it is useful to restate them in the paper rather than diluting our contribution. This has been a widely used practice in many papers. Even Theorem 1 in the TRPO paper is not new. We have clarified that these are established results. Taking a step back, even for these established results, we restate their proof (in the appendix not the main text) with our own language and notation system. The proof is very detailed and will be useful for readers who are not familiar with such knowledge. The amount of work and effort, not only the theoretical but also the empirical side, is substantial, given our analysis and extensive experiments. We understand the reviewer’s concern and are trying our best to respond and address them in a reasonable way. We also hope the reviewer could respect our effort rather than stating "copy and paste" because we use established theorems, which is a common practice, as the common philosophy of scientific research is “building on the shoulders of giants.”

---

> > ### Author Response · Authors · 2025-11-27
> > **Response to W3**
> >
> > First, we clarify that $p(o_t | s_t, a_t)$ is a conditional probability defined at a single time step. It represents the likelihood that a specific step $(s_t, a_t)$ originates from the expert policy, defined as $p(o_t=1 | s_t, a_t) \propto \exp(r(s_t, a_t))$. Defining it in this way is because the policy is defined as $\pi(a_t|s_t)$. As such, given a state and action pair, we are supposed to be able to calculate the probability of the action coming from the optimal policy. However, given that the optimal policy is unknown, we define the hidden variable $o_t$ so we can connect it with the reward function. This does not contradict the cumulative reward requirement. Since the optimality of a full trajectory depends on the joint probability of these events $p(o_{1:T} | \tau)$, maximizing the likelihood over the trajectory naturally results in maximizing the sum of rewards (cumulative reward), ensuring the policy is not myopic.

---

### Official Review · Reviewer_wZzT · 2025-10-25

**Soundness:** 3
**Presentation:** 2
**Contribution:** 1
**Rating:** 2
**Confidence:** 4

**Summary:**

This paper proposes an inverse RL based method for learning a process reward model for LLM reasoning. The method is based on maximum entropy IRL and an importance weighting method is proposed to approximate the partition function in the gradient. The authors showed connection to various methods such as DPO, DQO, etc. Experiments are conducted on several math and coding benchmarks, showing improvements on some. In the ablations, the authors show the effectiveness of their method for best-of-N sampling using an outcome reward model and training using only the learned process reward model.

**Strengths:**

* The proposed importance weighting method for the IRL algorithm is potentially a nice algorithmic trick.

**Weaknesses:**

* The connection with other SOTA method is a bit superfluous. The main commonalities is just that all methods are more or less based on maximum entropy RL, where the connections are widely known.
* IRL for LLM fine tuning has actually been proposed by multiple papers now e.g. [1, 2]. Not citing or comparing with them is missing a lot of context. Especially the proposed algorithm is very similar to [2].
* Considering the paper is algorithmic, it's missing some ablation experiments on the algorithmic design choices, for example using the importance weight vs taking more inner policy optimization steps. It is also widely known that this adversarial style IRL method can be unstable. How has that affected the proposed method, if at all?

[1] [Wulfmeier, Markus, et al. "Imitating language via scalable inverse reinforcement learning." Advances in Neural Information Processing Systems 37 (2024): 90714-90735.](https://arxiv.org/abs/2409.01369)

[2] [Li, J., Zeng, S., Wai, H. T., Li, C., Garcia, A., & Hong, M. (2024). Getting more juice out of the sft data: Reward learning from human demonstration improves sft for llm alignment. Advances in Neural Information Processing Systems, 37, 124292-124318.](https://arxiv.org/abs/2405.17888)

**Questions:**

* For AIME2024, why all baselines preformed significantly worse than the base model? Why rePIRL only matched the base model - should it ever surpass it?
* In Figure 1a, why should we expert PRM trained models to perform better on BON sampling?

---

> ### Author Response · Authors · 2025-11-24
> **Rebuttal (part-1)**
>
> >W1. The connection with other SOTA methods is widely known.
>
> A1. We respectfully disagree with the assessment that these relations are widely known. To the best of our knowledge, we are the first to provide a unified framework that grounds these baseline methods in rigorous theoretical derivation. Specifically, the connections to MathShepherd and PRIME are non-trivial (see Propositions 3 and 4, and Appendix B).
> Crucially, neither of these methods is inherently based on maximum entropy RL. Regarding MathShepherd, we demonstrate that our reward objective in Eq. (5) aligns with the cross-entropy loss used to train the PRM. Regarding PRIME, we show that it is a special case of our method that relies on strong assumptions which may not hold in real-world settings. These derivations are non-trivial and necessitate a  comprehensive grasp of both baseline methods and our method.
>
> >W2. Difference between rePIRL and existing IRL-based alignment method [1,2]
>
> A2. Thanks for raising these two works. We would like to kindly point out that these two methods are fundamentally different from rePIRL in the following aspects. (1) [1] relies on implicit rewards derived from the dataset via IQLearn and emphasizes the scalability of offline training, whereas rePIRL relies on online policy updates based on explicitly learned rewards from expert trajectories. [2] was designed for alignment rather than reasoning. As stated in [2], alignment mainly optimizes over the entire trajectories $(x, y)$, which fundamentally follows a bandit assumption (as discussed in our Section 3). However, reasoning requires modeling the problem as a real MDP, making the problem much more challenging. (2) To solve the challenge, we developed rePIRL under a token-level MDP and addressed the scalability (sample from unknown optimal policy) and intractability (partition function $Z$) issues with a series of new algorithmic designs. Our technical designs enable reliable recovery of an effective Process Reward Model (PRM) that provides token-level reward during LLM reasoning, leading to policies with stronger performance. (3) rePIRL conducted a rigorous theoretical analysis of major existing post-training methods and integrated them into the same framework with different assumptions. This analysis is helpful for obtaining a holistic understanding of RL-driven LLM post-training. As a result, rePIRL is the first work that generalizes classical IRL for LLM reasoning via a scalable learning algorithm. We believe this is a non-trivial contribution as there are many other papers accepted at top-tier ML conferences that generalize classical RL theorems or frameworks to scalable deep learning settings [1,2,3]. Will clarify the difference in the revised paper.
>
> >W3. Taking more policy inner iteration VS important sampling
>
> A3. First, we would like to clarify that our importance sampling is not heuristic; rather, it is grounded in rigorous theoretical derivations (see Lines 220–237). To further validate this, we compared our method against a baseline of simply increasing the number of inner policy iterations using the Qwen2.5-3B-Instruct model. Our method consistently outperforms this heuristic baseline across all math tasks. This empirical evidence confirms that the performance gains stem from the necessity of our importance sampling correction, rather than merely additional updates.
> |Model      | MATH-500 | AIME-2024 | Minerval_math | AMC  | Olympiadbench | Avg  |
> | ------------ | -------- | --------- | ------------- | ---- | ------------- | ---- |
> | rePIRL          | 62.4     | 10.0      | 27.2          | 38.5 | 29.3          | 33.5 |
> | Inner loop 3 | 59.0     | 10.0      | 27.9          | 36.1 | 29.3          | 32.4 |
> | Inner loop 5 | 61.6     | 3.3       | 24.3          | 37.3 | 28.4          | 30.9 |
>
> >W4. The adversarial style of IRL method can be unstable;
>
> A4. We acknowledge that traditional adversarial training is notoriously unstable, a phenomenon we also observed in our preliminary experiments. To mitigate this, we incorporated corrected trajectories generated by our policy into the reward model training, treating them as additional expert data. This effectively signals to the reward model (i.e., the discriminator) that these are high-quality trajectories, which in turn incentivizes the policy (i.e., the generator) to continue improving. Our ablation study confirms that this component is critical; including it significantly improves both the training stability and the final performance of our method.
>
> |Model | MATH-500 | AIME-2024 | Minerval_math | AMC  | Olympiadbench | Avg  |
> | ------- | -------- | --------- | ------------- | ---- | ------------- | ---- |
> | rePIRL     | 62.4     | 10.0      | 27.2          | 38.5 | 29.3          | 33.5 |
> | rePIRL w/o correct traj from policy | 61.4     | 3.3       | 31.6          | 34.9 | 27.6          | 31.7 |

---

> ### Author Response · Authors · 2025-11-24
> **Rebuttal (part-2)**
>
> >W5. Results on AIME 2024 dataset
>
> A5. It is important to note that AIME2024 is a harder task than benchmarks such as Math-500 and AMC [4, 5]; thus, achieving gains on this dataset is challenging. Furthermore, we conducted experiments using the Qwen3-4B-Base model, which is more powerful than Qwen2.5-3B-Instruct. Regarding AIME-2024, our method improved the RLOO baseline from 10% to 16%. This relative improvement of nearly 60% verifies the effectiveness of our approach.
> |Model                     | AMC   | MATH-500 | AIME-2024 | Minerval_math | Olympiadbench | Avg|
> | --------------------------- | ----- | -------- | --------- | ------------- | ------------- | ------------ |
> | Base model (Qwen3-4B-Base) | 36.1 | 58.6    | 6      | 19.9         | 32         | 30.5        |
> | RLOO                        | 39.7 | 73    | 10      | 29.8         | 39.7         | 38.4        |
> | rePIRL                    | 43.3 | 72.8    | 16      | 33.5         | 42.5         | 41.6        |
>
> >W6.  Why we should expect the PRM trained model perform better on BON Sampling
>
> A6. Thanks for pointing this out. Our model is a stronger policy than the base policy; as such, it can achieve a better performance upper bound than the weaker policy, which is reflected in the TTS experiment (under the same BoN experiment setting).
>
> References:
>
> [1] Learning Robust Rewards with Adversarial Inverse Reinforcement Learning, ICLR 2018
>
> [2]  Q♯: Provably Optimal Distributional RL for LLM Post-Training, Neurips 2025
>
> [3] PC-PG: Policy Cover Directed Exploration for Provable Policy Gradient Learning, Neurips 2020
>
> [4] RL Tango: Reinforcing Generator and Verifier Together for Language Reasoning, Neurips 2025
>
> [5] Qwen2.5-Math: The World's Leading Open-Sourced Mathematical LLMs. https://qwen.ai/blog?id=qwen2.5-math

---

> > ### Author Response · Authors · 2025-11-26
> >
> > Dear Reviewer,
> >
> > We sincerely appreciate your thoughtful feedback. As we enter the final week of the discussion period, we would be grateful to know whether our recent responses have addressed your concerns or if any questions remain. We are happy to provide any further clarification as needed. Wishing you a Happy Thanksgiving.
> >
> > Thank you,
> >
> > The Authors

---

### Official Review · Reviewer_tf3f · 2025-11-01

**Soundness:** 3
**Presentation:** 3
**Contribution:** 3
**Rating:** 6
**Confidence:** 2

**Summary:**

This paper introduces rePIRL, an inverse reinforcement learning (IRL) framework designed to learn process reward models (PRMs) for LLM reasoning. The authors contend that current PRM methods are limited by strong assumptions, such as requiring expert reward functions, or by intrinsic issues like entropy collapse. rePIRL addresses this by learning a PRM from expert trajectories using a dual learning process that interchangeably updates the policy and the PRM, utilizing custom techniques to scale traditional IRL to LLMs. The paper theoretically demonstrates that this framework can unify existing online and offline PRM learning methods, arguing that it operates with minimal assumptions. The method's effectiveness is validated empirically on math and coding datasets, where it reportedly outperforms existing PRM learning approaches.

**Strengths:**

1. The core contribution is the application of an IRL-inspired framework to learn PRMs. This is well-motivated, as it bypasses the need for explicit token-level reward annotations or access to an expert policy for MCTS, requiring only expert trajectories.

2. The paper provides a strong theoretical contribution by integrating several SOTA methods (DPO, DQO, MCTS, PRIME) into its framework as special cases that require additional assumptions. This analysis in Section 3.3 rigorously supports the central claim that rePIRL is a more general PRM learning method.

3. The method is tested on seven distinct math and coding benchmarks against relevant baselines, including BC, MCTS, and PRIME .

4. The ablations effectively validate the design choices. The "PRM-only" training experiment is particularly strong, demonstrating that the learned reward signal alone (without outcome rewards) can significantly improve performance over the base model, confirming the quality of the recovered reward function.

**Weaknesses:**

1. While rePIRL achieves SOTA average performance among the tested methods, the absolute improvements are modest. For example, on the math benchmarks, rePIRL achieves a 33.5% average, compared to 31.7% for vanilla RLOO and 30.7% for MCTS. These small margins raise questions about the practical utility of the method relative to its complexity.

2. The proposed dual learning algorithm is significantly more complex than the baselines. It requires simultaneously training a policy and a PRM , managing rollouts from both , and computing importance sampling weights. This complexity may present a high barrier to implementation and stable training.

3. The framework's "minimal assumptions" claim is traded for a strong dependency on high-quality expert trajectories. The experiments relied on generating four expert demonstrations per problem using Claude-3.7-sonnet, which is a powerful, proprietary model. This reliance on an expensive and extensive set of expert data is a significant practical limitation.

**Questions:**

none

---

> ### Author Response · Authors · 2025-11-24
> **Rebuttal**
>
> >W1. Justify the utility of rePIRL with respect to its complexity.
>
> A1. Thanks for raising this point. rePIRL delivers consistent improvements across all tasks without any task-specific reward labeling. To justify the statistical significance of the improvement, we conducted a paired t-test with null hypothesis H0 as Acc_{our} - Acc_{base} < 0. Our test results on three repeated runs all gave a p-value smaller than 0.05, showing the statistical significance. Further, we have added new ablations on Qwen3-4B base model, confirming that these improvements are robust across architectures. Such cross-model and cross-task stability is non-trivial for LLM reasoning, which is supported by our theoretical analysis.
> |Model                     | AMC   | MATH-500 | AIME-2024 | Minerval_math | Olympiadbench | Avg|
> | --------------------------- | ----- | -------- | --------- | ------------- | ------------- | ------------ |
> | Base model (qwen3-4B-Base) | 36.1 | 58.6    | 6.0      | 19.9         | 32.0         | 30.5        |
> | RLOO                        | 39.7 | 73.0    | 10.0      | 29.8         | 39.7         | 38.4        |
> | rePIRL                    | 43.3 | 72.8    | 16.0      | 33.5         | 42.5         | 41.6        |
>
> >W2. Justify the complexity of rePIRL.
>
> A2. Thanks for the comment. First, we would like to clarify that rePIRL only trains two models during learning, the reward model and the policy model (As we use RLOO for non-parametric value estimation, we don’t need to estimate the value function). This is the same as PPO or other methods that require learning the PRM reward model (PRIME and Math-Shepherd). As such, rePIRL does not introduce additional complexity compared to many SOTA methods. For example, on math tasks using Qwen2.5-3B-Instruct, the training time is 13.3 hours for rePIRL compared to 14.6 hours for PRIME. Besides, compared to these SOTA methods, rePIRL serves as a unified framework, where existing methods (DPO, DQO, Math-Shepherd, and PRIME) all arise as special cases with stronger approximations (Sec. 3.3). Empirically, rePIRL trains stably and outperforms all baselines consistently across tasks and model structures.
>
> >W3. rePIRL relies on expensive expert models.
>
> A3. We agree with the reviewer that rePIRL indeed needs expert trajectories with a certain quality to learn a reasonable reward model. However, existing SFT and PRM-based methods also need high-quality expert data. Even many RL training methods require an SFT warm-up phase, which also needs expert data. Here, we would like to point out that, first, rePIRL does not need extensive expert trajectories, i.e., 28K, which is way fewer than the training rollout during RL or the data needed for learning a good MCTS PRM. It is of a similar level as SFT warm-up, but achieves a way better performance than these methods. Second, we changed the Claude-3.7-sonnet (which is already not the latest Claude model) to an open-source model DeepSeek-R1-reasoning to collect the expert trajectories. The performance is comparable as shown in Figure 1 in our revised paper. The result further shows that rePIRL does not rely on expensive proprietary models to generate expert trajectories. In addition, rePIRL has the least assumption, i.e., it requires only sequence-level expert trajectories, not token-level reward supervision (required by supervised and DPO) or access to the expert policy or expert reward during training (required by MCTS). This makes rePIRL the least resource-intensive and assumption-minimal PRM learning framework.
>
> |Model             | MATH-500 | AIME-2024 | Minerval_math | AMC  | Olympiadbench | Avg  |
> | ------------------- | -------- | --------- | ------------- | ---- | ------------- | ---- |
> | rePIRL Claude 3.7   | 62.4     | 10.0      | 27.2          | 38.5 | 29.3          | 33.5 |
> | rePIRL Deep-seek-r1 | 61.5     | 10.0      | 28.1          | 37.6 | 29.0          | 33.2 |

---

> > ### Comment · Reviewer_tf3f · 2025-11-24
> >
> > Thank you for your reply. I believe my questions have been resolved, so I have decided to increase my rating.

---

> > > ### Author Response · Authors · 2025-11-24
> > > **Thank you!**
> > >
> > > Thank you for your constructive feedback and for taking the time to review our rebuttal. We appreciate your increased rating and are glad that we could address your concerns satisfactorily.

---

### Official Review · Reviewer_ZTdJ · 2025-11-01

**Soundness:** 3
**Presentation:** 3
**Contribution:** 3
**Rating:** 6
**Confidence:** 3

**Summary:**

This paper introduces rePIRL, a novel framework for learning Process Reward Models (PRMs) in large language models (LLMs) for multi-step reasoning tasks. The approach centers on adapting Inverse Reinforcement Learning (IRL) to recover token-level reward signals from expert trajectories, without requiring step-wise annotations, preference labels, or access to the expert policy itself. The method alternates between updating a policy (via maximum entropy RL) and a reward model, employing importance sampling to render the IRL objective tractable for LLM-scale problems. Theoretically, the authors demonstrate that several existing PRM learning methods (e.g., PRIME, Math-Shepherd, DPO, DQO) can be viewed as special cases of their framework under more restrictive assumptions, thereby positioning rePIRL as a more general approach. Empirically, rePIRL outperforms baselines on both mathematical (AIME, AMC, Math-500) and coding (Leetcode, LiveCodeBench) benchmarks when fine-tuning a Qwen2.5-3B model.

**Strengths:**

The paper makes a substantial contribution by successfully adapting and scaling classical IRL to the challenging domain of LLM reasoning. While IRL is well-established in robotics and control, its application to LLMs is novel and non-trivial due to the enormous state and action spaces involved. The theoretical unification of existing online and offline PRM methods under a single framework is both creative and insightful, clearly underscoring the minimal-assumption nature of rePIRL. The technical execution is sound: the derivation of the learning objective—including the use of importance sampling to circumvent the intractable partition function—is rigorous, and comprehensive theoretical proofs (provided in the appendix) rigorously connect rePIRL to prior work. The experimental evaluation is thorough, spanning multiple challenging benchmarks and including ablations on policy update methods and test-time scaling. The results consistently demonstrate superior performance compared to strong baselines.

**Weaknesses:**

While the paper includes several important baselines, the empirical comparison could be further strengthened by incorporating a broader array of recent PRM or preference-based methods, particularly those that similarly operate under minimal supervision. Although the paper ablates the policy learning algorithm, it offers less insight into the PRM's architecture and specific design choices. Given that the reward model is a core component, an ablation study of its capacity or architectural variations would be valuable for understanding its impact on final performance and training stability.

**Questions:**

Please refer to "Weaknesses".

---

> ### Author Response · Authors · 2025-11-24
> **Rebuttal**
>
> Thanks to the reviewer for the insightful comments. Following the reviewer’s suggestion, we further included a newer PRM method, RL Tango, and a newer preference-based method: KTO. Note that our setting does not naturally support KTO, which requires preference labels. Our expert dataset does not require preference labels. To run KTO, we took extra effort to add preference labels for KTO. The results can be found in Table 2 of our revised paper.
> | Model | MATH500 | AIME2024 | MinervaMath | AMC | OlympiadBench | Avg |
> | :--- | :--- | :--- | :--- | :--- | :--- | :--- |
> | Qwen2.5-3B-Instruct | 46.0 | 10.0 | 22.4 | 34.9 | 28.9 | 28.4 |
> | BC | 52.0 | 3.3 | 8.8 | 22.8 | 15.9 | 20.6 |
> | KTO | 54.5 | 4.8 | 18.5 | 29.5 | 24.2 | 26.3 |
> | RL-Tango | 57.1 | 6.6 | 25.2 | 32.5 | 29.5 | 30.2 |
> | MCTS | 58.0 | 6.6 | 24.3 | 33.7 | 30.7 | 30.7 |
> | PRIME | 56.2 | 6.6 | 26.5 | 31.3 | 29.0 | 29.9 |
> | RLOO | 63.6 | 3.3 | 26.1 | 36.1 | 29.6 | 31.7 |
> | **rePIRL** | 62.4 | 10.0 | 27.2 | 38.5 | 29.3 | **33.5** |
>
>
> Second, we conducted additional ablation studies on the model’s architecture. The current rePIRL results utilize PRM initialized from the same base language model as the policy model (Qwen2.5-3B-Instruct). We changed the reward model sizes and architectures as follows.
> |Model                           | MATH-500 | AIME-2024 | Minerval_math | AMC  | Olympiadbench | Avg  |
> | --------------------------------- | -------- | --------- | ------------- | ---- | ------------- | ---- |
> | 3B instruct for reward modeling   | 62.4     | 10.0      | 27.2          | 38.5 | 29.3          | 33.5 |
> | 1.5B instruct for reward modeling | 57.80    | 10.0      | 27.6          | 36.1 | 27.6          | 31.8 |
>
> The complete experimental results can also be found in Appendix D.1 of the revised paper. We observe that replacing the reward model with a smaller one degrades performance. Nevertheless, our approach still outperforms the RLOO baselines, demonstrating that rePIRL generalizes across different reward model architectures and sizes.
>
> We would like to further clarify that it is a common practice to use Qwen models, the best-performed open-source model family for experiments and ablation, given that different Qwen models have different structures. We do not consider other LLM model families, e.g., LLama and non-LLMs, given the performance difference of their base models.

---

> ### Author Response · Authors · 2025-11-26
>
> Dear Reviewer,
>
> We sincerely appreciate your thoughtful feedback. As we enter the final week of the discussion period, we would be grateful to know whether our recent responses have addressed your concerns or if any questions remain. We are happy to provide any further clarification as needed. Wishing you a Happy Thanksgiving.
>
> Thank you,
>
> The Authors

---

### Author Response · Authors · 2025-11-24
**Summary of response**

We thank the reviewers for the constructive feedback. Below, we summarize our responses:

**New experiments**

We added all experiments suggested by reviewers and updated the paper accordingly. Below, we give a brief summary.

1. We applied rePIRL to the newest Qwen-3-4B model and demonstrated its superiority over baselines on this new model, which further justifying the generalizability of our method across different base models. The experiment was added to Sec 4.2 in Table 2.

2. According to reviewer ZTdJ’s comment, we added two more baselines: RL Tango (PRM method); KTO (preference learning). The results were shown in Table 2, which demonstrates our advantage.

3. Following reviewer ZTdJ’s comment, we added an ablation study to show that rePIRL is generalizable across different reward model architectures and sizes. The experiment was added to Sec 4.3 and Appendix D.1.

4. In responding to reviewer tf3f’s concern on expert trajectory cost, we used an open-source model (DeepSeek-R1-reasoning) as our expert model and demonstrated the effectiveness of rePIRL under this expert model. We added this experiment to Sec 4.3.

5. We added ablation studies on (1) our rollout strategy to ensure training stability; (2) sampling vs taking more inner loop for reward learning to address reviewer wZzT’s concerns. We added this experiment to Sec 5 and Sec 4.3.

Besides, we also updated the paper to clarify the following key concerns.
We updated the introduction to clarify and emphasize our technical contributions, especially on the theoretical side.
We updated Section 3 to clarify that Theorem 1 is our theoretical foundation rather than novelty; Our novelty is the analysis of different methods’ connection to our framework and their assumptions. (In response to reviewer TGya)
We updated the related work to discuss the methods pointed out by reviewer TGya.
We added a discussion on the complexity of rePIRL in response to reviewer tf3f’s comment.

We responded to **all the comments** from reviewers in our rebuttal. Below, we summarized the key points.

Reviewer ZTdJ

We followed the reviewer’s comment to add additional baselines and required ablation studies. The results further show the advantage of our method over existing PRM and preference learning methods. They also demonstrate the robustness of our method against reward model architectures and sizes.

Reviewer tf3f

We clarified that our method maintains a similar level of complexity compared to many major SOTA post-training methods (e.g., PRIME and PRM methods). We also showed the run-time comparison with PRIME as a quantitative justification. We further justified that our method does not need crazy expert trajectories (similar level to many existing post-training methods). It can also achieve performance gain over existing methods when using an open-source model as the expert, addressing the concerns on relying on expensive expert trajectories.

Reviewer wZzT

We first further emphasized our technical contribution and discussed the difference between rePIRL and existing works. We then conducted additional ablation studies to respond to the reviewer’s comment on training iteration \& sampling and training stability. Finally, we explain the performance gain on AIME 2024 and testing-time scaling.

Reviewer TGya

We discussed the fundamental difference between rePIRL and the related works pointed out by the reviewer. We also clarified the theoretical contribution. Finally, we explained a notation confusion pointed out by the reviewer and further justified that our learning objective indeed considered long-term cumulative reward.

---

### Meta-Review · Area_Chair_rFU4 · 2026-01-02

**Summary:**

This paper proposes an inverse RL-inspired framework, which adopts a dual learning process that updates the policy and the PRM interchangeably, to improve upon the traditional PRM methods relying on strong assumptions about expert policies or suffering from issues like entropy collapse.

The major weaknesses raised by reviewers include: 1) unconvincing technical novelty and contribution compared with existing studies, particularly the paper “Getting More Juice Out of the SFT Data: Reward Learning from Human Demonstration Improves SFT for LLM Alignment” published in NeurIPS 2024; 2) missing some important ablation experiments about specific design choices; 3) modest performance improvements; 4)  complexity and reliance on an expensive and extensive set of expert data; 5) lack of some discussions and clarifications.

**Reviewer Concerns:**

During rebuttal, the authors have addressed some of the reviewers’ concerns. However, the major weakness (1) - technical similarity to existing work, which was a consensus reached by both Reviewer wZzT and Reviewer TGya, has not been totally resolved. Although the authors provided clarifications on the difference in the rebuttal, Reviewer TGya still questioned on the technical challenge and difference. The AC also thinks that this issue needs to be fixed through more in-depth discussions and analyses to improve the quality of this work.

**Reviewer Scores:**

The final socres would be 2, 2, 6, 8.

---

### Decision · Program_Chairs · 2026-01-26

Reject